# The Potential Habitat Response of *Cyclobalanopsis gilva* to Climate Change

**DOI:** 10.3390/plants13162336

**Published:** 2024-08-22

**Authors:** Bao Liu, Yinglin Li, Jintao Zhao, Huiying Weng, Xingzhuang Ye, Shouqun Liu, Zixin Zhao, Sagheer Ahmad, Chaoyu Zhan

**Affiliations:** 1Forestry College, Fujian Agriculture and Forestry University, Fuzhou 350002, China; fafuwhy@163.com (H.W.); yxz@fafu.edu.cn (X.Y.); 15131902213@163.com (S.L.); 19914726542@163.com (Z.Z.); 15129774472@163.com (C.Z.); 2Hunan Yiyang City Heshan District Forestry Bureau, Yiyang 413000, China; liyinglin2015@126.com; 3College of JunCao Science and Ecology, Fujian Agriculture and Forestry University, Fuzhou 350002, China; fafuzjt@163.com; 4College of Landscape Architecture and Art, Fujian Agriculture and Forestry University, Fuzhou 350002, China; sagheerhortii@gmail.com

**Keywords:** *Cyclobalanopsis gilva*, MaxEnt model, climate change, suitable habitat

## Abstract

*Cyclobalanopsis gilva*, a valuable timber species in China, holds significant importance for understanding the constraints imposed by climate change on the dynamic geographic distribution of tree species. This study utilized the MaxEnt maximum entropy model to reconstruct the migratory dynamics of *C. gilva* geographical distribution since the Last Glacial Maximum. The objective was to comprehend the restrictive mechanisms of environmental factors on its potential geographical distribution, aiming to provide insights for mid-to-long-term afforestation planning of *C. gilva*. The optimized MaxEnt model exhibited a significantly high predictive accuracy, with an average AUC value of 0.949 ± 0.004 for the modern suitable habitat model of *C. gilva*. The total suitable habitat area for *C. gilva* in contemporary times was 143.05 × 10^4^ km^2^, with a highly suitable habitat area of 3.14 × 10^4^ km^2^. The contemporary suitable habitat was primarily located in the southeastern regions of China, while the highly suitable habitat was concentrated in eastern Fujian and central-eastern Taiwan. Bioclimatic variables such as mean diurnal range (Bio2), min temperature of coldest month (Bio6), precipitation of driest quarter (Bio17), and precipitation of driest month (Bio14) predominantly influenced the modern geographic distribution pattern of *C. gilva*, with temperature factors playing a leading role. With global climate warming, there is a risk of fragmentation or even loss of suitable habitat for *C. gilva* by 2050 and 2090. Therefore, the findings of this study can significantly contribute to initiating a habitat conservation campaign for this species.

## 1. Introduction

Climate change is a crucial driver influencing the patterns of species distribution, with significant impacts on the geographic distribution patterns and genetic structures of modern biota since the Quaternary, particularly during interglacial and glacial oscillations, such as the Last Glacial Maximum [1,2,3]. In recent years, the effects of climate warming on plant distribution have become a focal point in botanical, ecological, and interdisciplinary research. Studies indicate that many plants respond to climate warming by altering their geographical distribution, for instance, by migrating towards higher latitudes or elevations [4]. Research suggests that the Earth is currently facing the sixth mass extinction due to climate warming since the 20th century, and it is occurring at a faster rate than the previous five mass extinctions in geological history [5]. The risk of extinction for endangered species is expected to significantly increase with ongoing global climate warming [6]. Thus, understanding the changing trends in plant distribution patterns under global climate warming and revealing their response mechanisms to climate change is of paramount importance for developing rational conservation strategies and effective biodiversity conservation measures.

In recent years, species distribution models (SDMs) have been widely employed to analyze the impact of climate change on species distribution [7,8]. By inputting climate data for different contemporary and future climate change scenarios into models, it is possible to predict changes in suitable habitat for species under different future climate scenarios and explore the restrictive mechanisms of bioclimatic variables on the geographic distribution of species [9]. With technological advancements, various species distribution models have emerged, including Bioclim models [10], genetic algorithm for rule set production (GARP) models [11], DOMAIN models [12], and maximum entropy models (MaxEnt) [13]. Due to its fast computation speed, simplicity, and the ability to maintain high predictive accuracy even with limited sample data, the MaxEnt model is commonly used for predicting species potential distribution [14]. Currently, the MaxEnt model has been widely applied in various research fields, such as predicting the potential geographic distribution of invasive species [15], endangered species [16], and pest control [17], contributing significantly to research in conservation biology, global change biology, biogeography, and ecology [18].

*Cyclobalanopsis gilva*, also known as Chinese ring-cupped oak, is an evergreen tree species belonging to the Fagaceae family. It is a valuable timber species with excellent wood texture, resistance to pests, and durability in wet conditions, making it one of the excellent hardwoods used in the production of high-end furniture, vehicles, textile equipment, musical instruments, and sports equipment. *C. gilva* is mainly distributed in provinces such as Zhejiang, Fujian, Hunan, Guangdong, Jiangxi, and Guizhou in China [19]. It exhibits strong adaptability to the environment, is relatively drought-resistant, and possesses ecological functions such as water source conservation and soil retention, making it a highly valuable tree species for landscaping [20]. Moreover, the intensification of global climate warming and the increasing demand for precious hardwood timber have made the population renewal of *C. gilva* more challenging. Due to the high demand for *C. gilva* as a valuable hardwood, it has been overexploited for an extended period, leading to scarce existing natural resources [21]. Therefore, urgent measures are needed for the conservation of *C. gilva*. Current research on *C. gilva* mainly focuses on proteomics analysis [22], community structure [23], and seedling breeding technology [24]. Some scholars have conducted research on the potential geographical distribution of *C. gilva*. They explored the changes in the potential distribution patterns of *C. gilva* under different future climate conditions, revealing varying degrees of expansion in potential suitable habitats in four scenarios for two future periods. Additionally, a general trend of migration towards higher latitudes was observed [25]. However, the dynamic migration pattern of potential suitable habitats for *C. gilva* since the Last Interglacial Period and the main bioclimatic variables constraining its potential geographic distribution remain unclear. This knowledge gap hinders the effective conservation and utilization of *C. gilva* germplasm resources.

Based on field surveys, this study applied the ENMeval package to assist in selecting the MaxEnt model to predict the potential distribution patterns of *C. gilva* under past, present, and future climate change scenarios. Through this research, we aim to address the following questions: (1) What is the current potential geographic distribution of *C. gilva*? (2) How has the distribution pattern of *C. gilva* migrated under past and future climate change scenarios? (3) What are the main bioclimatic variables constraining the potential geographic distribution of *C. gilva*? (4) What are the restrictive mechanisms of these bioclimatic variables on the potential geographic distribution of *C. gilva*? Our study will provide valuable references for the conservation and utilization of germplasm resources for *C. gilva* and other valuable hardwood tree species in the Fagaceae family.

## 2. Materials and Methods

### 2.1. Data Collection for Cyclobalanopsis gilva Distribution Points

Geographic distribution point data for *C. gilva* were primarily sourced from three main platforms: the China Teaching Specimen Resource Sharing Platform (http://mnh.scu.edu.cn/, last accessed on 20 March 2024), the Chinese Academy of Sciences Natural Specimen Museum (http://www.cfh.ac.cn/, last accessed on 20 March 2024), and the Global Biodiversity Information Facility (https://www.gbif.org/, last accessed on 20 March 2024). These platforms served as the primary references, supplemented with the relevant literature documenting the actual geographical distribution of *C. gilva*. The observed distribution data points of *C. gilva* span from 1980 to 2023. For distribution points lacking latitude and longitude information, GPS coordinates were extracted using GPS positioning systems.

A total of 190 geographic distribution points for *C. gilva* were obtained. Subsequently, the distribution data underwent a screening process, removing duplicates, points with ambiguous location records, and those with outdated temporal information. Using ArcGIS 10.8 with a 10 km × 10 km buffer zone, the filtering process was performed based on the principle of retaining only one distribution point within intersecting buffer zones to minimize biases caused by clustering effects. This resulted in a final set of 73 naturally occurring distribution points for *C. gilva* (Figure 1). The remaining distribution points, post-screening, were selected, and their latitude and longitude coordinates were extracted and saved in “.csv” format for further use. The geographical distribution data utilized in this study were primarily sourced from recent records of herbaria, each confirming the presence of at least one individual of *C. gilva* at specific locations. These data were categorical, indicating presence/absence rather than providing quantitative measures of abundance or density.

### 2.2. Source and Filter of Bioclimatic Variables

Climate data for 19 climate variable factors were obtained from the WorldClim database (http://worldclim.org, last accessed on 25 March 2024). To enhance model accuracy, climate data were selected for six different periods: the Last Interglacial Period, Last Glacial Maximum, mid-Holocene, Contemporary (1970–2000), and future scenarios for the 2050s (2041–2060) and 2090s (2081–2100). The future scenarios for the 2050s and 2090s included both low greenhouse gas emission (ssp126) and high emission (ssp585) climate scenarios.

Climate data for the modern and future periods were generated using the second-generation National Climate Center medium resolution climate system model (BCC-CSM2-MR). For the Last Interglacial Period, Last Glacial Maximum, and mid-Holocene, data from the community climate system model version 4 (CCSM4) were used due to the absence of BCC-CSM2-MR data. The spatial resolution for all periods, except for the Last Glacial Maximum (LGM) which had a rate of 2.5 arc-minutes, was uniformly 30 arc-seconds. 

To address collinearity among climate variables and avoid overfitting, a pre-modeling experiment was conducted. The 19 climate variables were pre-simulated with 10 repetitions, excluding variables with a contribution rate of 0. Subsequently, ArcGIS 10.8 was employed to sample the bioclimatic variable data at the 73 naturally occurring distribution points. SPSS 26.0 was used to conduct Pearson’s multicollinearity analysis on the 19 bioclimatic variables. Variables with an absolute correlation coefficient |r| > 0.8 were selected, prioritizing those with significant contributions and close associations with *C. gilva* geographical distribution. Eight bioclimatic variables were ultimately selected: annual mean temperature (Bio1), mean diurnal range (Bio2), max temperature of warmest month (Bio5), min temperature of coldest month (Bio6), mean temperature of wettest quarter (Bio8), precipitation of driest month (Bio14), precipitation seasonality (coefficient of variation) (Bio15), and precipitation of driest quarter (Bio17).

### 2.3. Model Optimization

The ENMeval package (R4.1.1) was utilized to assist in optimizing the parameters of the MaxEnt model. The regularization multiplier (RM) and feature combinations (FC) were crucial parameters in the MaxEnt species distribution model. RM was set from 0.1 to 4, incrementing by 0.1 in each step, resulting in 40 RM parameters. Eight feature combinations (L, LQ, H, LQH, LQHP, LQHPT, HPT, QPT) were employed. Here, L represents linear, Q represents quadratic, H represents hinge, P represents product, and T represents threshold. The other parameters were kept at default settings. ENMeval was employed to test the 320 parameter combinations using delta.AICc to assess the model’s fitness and complexity, along with avg. diff. AUC and avg. test. or10pct to examine overfitting between species distribution points. The parameter combination with delta.AICc = 0 was selected as the optimal parameter set for application in MaxEnt model building.

### 2.4. Model Establishment and Evaluation

The filtered species distribution point data (in .csv format) and bioclimatic variable data (in .asc format) were imported into MaxEnt 3.4.1 (https://biodiversityinformatics.amnh.org/open_source/maxent/, last accessed on 15 February 2024). The bootstrap method was selected for sampling, with 75% of species distribution data randomly chosen as the training set and the remaining 25% as the testing set. The RM and FC parameters optimized by the ENMeval package were used. The model was run 10 times to mitigate the interference of outliers on the model results, with other parameters kept at default settings. The jackknife method was used to evaluate the influence of each bioclimatic variable on *C. gilva*, and results were generated in logistic and ASCII format.

The receiver operating characteristic (ROC) curve, plotting false positive rate (FPR) against true positive rate (TPR), was utilized for model evaluation. FPR represents the probability of predicting positive when there is no actual species distribution, while TPR represents the probability of predicting positive when there is an actual species distribution. The area under the ROC curve (AUC) was used to assess the accuracy of the model predictions. AUC values close to 1 indicated a strong correlation between bioclimatic variables and the predicted geographical distribution of the species, reflecting higher predictive accuracy. Generally, AUC values less than 0.6 indicated prediction failure, 0.6–0.7 suggested poor predictions, 0.7–0.8 indicated moderate predictions, 0.8–0.9 represented good predictions, and 0.9–1.0 indicated excellent predictions.

### 2.5. Habitat Suitability Level Analysis and Area Statistics

The average results of the 10 repetitions of the *C. gilva* model were imported into ArcGIS 10.8. Using the conversion tool, the ASCII files were transformed into raster files. Based on the “manual grading method”, suitable habitats for *C. gilva* were classified into four distinct levels, as follows, using a reclassification tool: inhospitable habitat (0–0.2), low suitable habitat (0.2–0.5), moderate suitable habitat (0.5–0.7), and highly suitable habitat (0.7–1). The final distribution map of *C. gilva* in different climates and scenarios was generated, and the habitat area for different periods and scenarios was statistically analyzed.

### 2.6. Spatial Distribution Pattern Changes in Suitable Habitat

The predictive outcomes were reclassified such that areas with a species’ probability of presence below 0.5 were categorized as unsuitable habitats, and were assigned the value “0”; conversely, areas where the probability of presence met or exceeded 0.5 were identified as suitable habitats, and were assigned the value “1”. The established matrices (0, 1) indicated the presence or absence of *C. gilva* geographical distribution under different climate scenarios in the past, present, and future. The temporal slice of the modern climate, extending from 1970 to 2000, was established as a reference benchmark for gauging shifts in the distributional range of suitable habitats for *C. gilva*. This epoch served as a foundational baseline, against which habitat dynamics—including expansions, contractions, and consistencies—were quantified within the context of diverse climatic projections. Changes in suitable habitat area for different suitability levels were calculated, distinguishing between newly gained, retained, or lost suitable habitat. Using the ArcGIS 10.8, the centroids of *C. gilva* for different periods and climate scenarios were analyzed, providing insights into the dynamic changes in suitable habitat over time and under varying climate scenarios. The specific method involved identifying spatial units with distribution probability values that met certain criteria as suitable habitats, followed by calculating the geometric center of these units, which was the average of all suitable unit coordinates. This process yielded the centroid coordinates that represented the overall spatial location of the species’ suitable habitat.

## 3. Results and Analysis

### 3.1. Model Optimization and Accuracy Evaluation

The MaxEnt model is a complex machine learning model. Its default parameters are derived from testing 266 species in different regions, demonstrating good performance in predicting actual species distributions. However, the default MaxEnt model may exhibit poor transferability, leading to overfitting and significant prediction errors in potential species distribution. Therefore, the optimization of MaxEnt model parameters was necessary. Using the ENMeval package based on 73 distribution points of *C. gilva*, the MaxEnt model was optimized.

Table 1 shows that when the feature combination was FC = LQ and the regularization multiplier was RM = 1.8, the delta.AICc = 0. In contrast, simulating with the default MaxEnt parameters (FC = LQHPT, RM = 1) resulted in delta.AICc = 80.564. Moreover, the optimized model’s avg. diff. AUC and avg. test. or10pct were lower than those of the default MaxEnt model, reducing by approximately 51% and 7%, respectively. This indicated that the optimized parameters significantly reduced model overfitting, making the optimized model the best choice.

In the optimized MaxEnt model, habitat suitability for *C. gilva* was predicted. The ROC curve of the model (Figure 2) showed an average training AUC value of 0.949 with a mean standard deviation of 0.004 over 10 repetitions, indicating excellent prediction results (>0.9).

### 3.2. Importance of Bioclimatic Variables

As indicated in Table 2,examining the contribution rates, the most crucial bioclimatic variables collectively accounted for 90.6%. These factors, along with their respective suitable ranges, included mean diurnal range (Bio2, 40%), with a suitable range of 3.92~9.30 °C; min temperature of coldest month (Bio6, 28.1%), with a suitable range of 5.15~15.05 °C; precipitation of driest quarter (Bio17, 16.9%), with a suitable range of 75.30~577.40 mm; coefficient of precipitation seasonality (coefficient of variation) (Bio15, 5.6%), with a suitable range of 10.99~79.43 mm; and precipitation of driest month (Bio14, 4.9%), with a suitable range of 20.08~173.70 mm. The top five factors by importance contributed to a cumulative total of 97.4%, comprising min temperature of coldest month (Bio6, 54.1%), mean diurnal range (Bio2, 19.7%), precipitation seasonality (coefficient of variation) (Bio15, 12.8%), mean temperature of wettest quarter (Bio8, 8.7%), and precipitation of driest month (Bio14, 2.1%).

Table 2 illustrates that, when using individual variables, the top three variables with the highest regularization training gain were mean diurnal range (Bio2), precipitation of driest quarter (Bio17), and precipitation of driest month (Bio14). Similarly, in testing, the highest gain came from precipitation of driest month (Bio14), mean diurnal range (Bio2), and precipitation of driest quarter (Bio17). The top three variables for AUC were precipitation of driest month (Bio14), precipitation of driest quarter (Bio17), and mean diurnal range (Bio2). This suggests that these bioclimatic variables encompassed more valuable information.

When using variables other than the aforementioned, the top three variables with the most reduced regularization training gain were annual mean temperature (Bio1), max temperature of warmest month (Bio5), and precipitation of driest month (Bio14). In testing, the highest reduction came from precipitation of driest quarter (Bio17), precipitation of driest month (Bio14), and annual mean temperature (Bio1). The three variables with the most reduced AUC values were precipitation of driest quarter (Bio17), precipitation of driest month (Bio14), and precipitation seasonality (coefficient of variation) (Bio15). This indicated that these variables contained unique information that was not present in other bioclimatic variables.

In summary, the primary bioclimatic variables influencing the modern geographic distribution of *C. gilva* were temperature-related (mean diurnal range(Bio2), min temperature of coldest month(Bio6), annual mean temperature(Bio1), max temperature of warmest month(Bio5), mean temperature of wettest quarter(Bio8)) and precipitation-related (precipitation of driest month (Bio14), precipitation seasonality (coefficient of variation) (Bio15), precipitation of driest quarter (Bio17)).

### 3.3. Modern Potential Distribution

Among the 73 modern distribution records of *C. gilva*, the proportions for highly suitable, moderately suitable, and generally suitable areas were 12.33%, 39.73%, and 45.21%, respectively. As indicated in Table 3, the total area of the modern suitable region for *C. gilva* was 143.05 × 10^4^ km^2^, with the highly suitable region covering 3.14 × 10^4^ km^2^. Figure 3d illustrates that the suitable distribution areas mainly included regions in Fujian, Zhejiang, southern Jiangsu, southern Anhui, southern Hubei, Hunan, Chongqing, most parts of Guizhou, large portions of Guangxi, Guangdong, most parts of Hainan, and large portions of Taiwan. The highly suitable region was mainly concentrated in central and eastern Taiwan, partial eastern Fujian, and southern Hunan.

The average suitability of the 73 species distribution points was 0.50. The point with the highest suitability was Miaoli County in Taiwan (0.86), while the lowest suitability was observed in Changhua County, Taiwan (0.11).

### 3.4. Past and Future Potential Distribution

From Figure 4 and Figure 5, it can be observed that the suitable distribution areas of *C. gilva* underwent significant changes in different periods and under different climate scenarios. As indicated in Table 3 and Table 4, during the Last Interglacial Period (LIG), the total suitable area (146.68 × 10^4^ km^2^) was slightly larger than the modern era, with a 2.54% increase, and the highly suitable area (3.87 × 10^4^ km^2^) significantly expanded by 23.25%. This highly suitable area was primarily distributed in the eastern parts of Fujian and Zhejiang, and central and eastern Taiwan. In the Last Glacial Maximum, both the total suitable area and the highly suitable area reached their maximum values at 345.72 × 10^4^ km^2^ and 46.52 × 10^4^ km^2^, respectively, showing a 141.68% and 1381.53% increase, respectively, compared with the modern era. Notably, the highly suitable area in Taiwan transitioned from highly suitable in other periods to moderate suitable during this period. During this time, the East China Sea shelf evolved into land, becoming a moderate to low suitable area for *C. gilva*. In the mid-Holocene, the potential suitable distribution of *C. gilva* was similar to the present, with the highly suitable area mainly in central and eastern Taiwan. The total suitable area and highly suitable area were 147.67 × 10^4^ km^2^ and 2.92 × 10^4^ km^2^, respectively.

Comparing the future scenarios with the modern era, except for 2090 s-ssp585, the total suitable area of *C. gilva* increased in the future scenarios. However, the highly suitable area significantly contracted, mainly remaining in the central and eastern parts of Taiwan. The growth in the total suitable area was mainly attributed to the expansion of the moderately suitable area, especially the low suitable area. In the 2050 s-ssp126 scenario, the highly suitable area of *C. gilva*, compared with the modern era, was reduced to only the central and eastern parts of Taiwan. Although the total suitable area increased by 16.76% compared with the modern era, the highly suitable area contracted by 59.55%. In the ssp126 scenario, in the 2090s, compared with the 2050s, although the total suitable area shrunk by 3.85%, the highly suitable area expanded by 30.71%. In the ssp585 scenario, by the 2090s, the highly suitable area of *C. gilva* drastically shrunk, remaining only at 1.02 × 10^4^ km^2^, a 67.52% reduction compared with the modern era. The moderately suitable area also stopped expanding and remained similar in size to the modern era. Compared with the other three scenarios, the growth rate in this scenario significantly decreased, and the loss rate increased to some extent, resulting in a negative change rate of −3.65% in 2090 s-ssp585 (Table 4).

### 3.5. Centroid Migration Routes

As shown in Figure 6, during the Last Glacial Maximum, the centroid of the suitable area for *C. gilva* was in Zhongcun Township, Yongfeng County, Ji’an City, Jiangxi Province. By the Last Glacial Maximum, the centroid had migrated northwest to Tongzhong Township, Chongyang County, Xianning City, Hubei Province, a distance of 331.68 km from the Last Glacial Maximum. By the mid-Holocene, the centroid of the suitable area for *C. gilva* migrated southwest to Xingqiao Town, Jingzhou District, Ji’an City, Jiangxi Province, a distance of 263.67 km from the Last Glacial Maximum. In the modern era, the centroid of the suitable area for *C. gilva* was in Fengtian Town, Anfu County, Ji’an City, Jiangxi Province, which had shifted northward by 28.95 km compared with the mid-Holocene.

In the ssp126 climate scenario, from the modern era to the 2050s and then to the 2090s, the centroid of the suitable area for *C. gilva* first shifted northeast by approximately 36.22 km to Jinjiang Township, Xiajiang County, Ji’an City, Jiangxi Province. It then shifted southeast by about 62.01 km to Tangcheng Township, Yongfeng County, Ji’an City, Jiangxi Province. Meanwhile, in the ssp585 climate scenario, from the modern era to the 2050s and then to the 2090s, the centroid of the suitable area for *C. gilva* first shifted northeast by about 93.52 km to Daifang Town, Le’an County, Fuzhou City, Jiangxi Province. It then shifted southwest by approximately 127.05 km to Tianhe Town, Ji’an County, Ji’an City, Jiangxi Province. Overall, there was a tendency for the future suitable area of *C. gilva* to migrate toward higher latitudes, but the northward expansion was not significant.

## 4. Discussion

### 4.1. Overall Assessment of Prediction Results

In this study, the ENMeval package was employed to optimize the MaxEnt model parameters, selecting regularization parameters and feature combinations with the minimum AICc and larger AUC values. The optimized parameters used in this study were FC = LQ, RM = 1.8. When these optimized parameters were applied, the AUC values of the MaxEnt model for *C. gilva* under each climate scenario exceeded 0.9, indicating high predictive accuracy. The predicted suitable areas covered all distribution points of *C. gilva*, and the simulated modern distribution closely matched its actual distribution points. These results suggest the reliability of the predictions in this study.

The uniformity of sample collection points and the coverage of samples played a crucial role in model accuracy. Proximity to the true distribution state of the species and a broad sampling range contributed to higher predictive accuracy. In this study, the distribution sources of *C. gilva* covered multiple provincial-level administrative regions, including Hubei, Hunan, Jiangxi, Guizhou, Guangdong, Zhejiang, Taiwan, and Fujian. This broad sampling range enhanced the accuracy of the model predictions. Studies have indicated that overly dense geographical coordinates of species distribution may lead to model complexity, reducing transferability and, consequently, lowering predictive accuracy [26]. In this study, spatial dilution of geographic population information for specimens within a grid cell was performed, with distribution point data imported into ArcGIS using a 10 km buffer distance. The principle of retaining only one distribution point within intersecting buffers was applied to reduce biases caused by clustering effects, thereby improving predictive accuracy. Concurrently, the modern climate data utilized in our research were sourced from the 2.1 version of the WorldClim database, which was temporally bound to the period between 1970 and 2000, and did not include climate information for the years ranging from 1800 to 1969. This limitation in the data’s temporal range presented a potential constraint on our study.

### 4.2. Constraints of Climatic Factors on the Geographic Distribution of Cyclobalanopsis gilva

The geographic distribution of plants is the result of the combined effects of biotic and abiotic factors, reflecting the ultimate response of plants to the environment. The main factors influencing the geographic distribution of plants include climate [27], topography [28], and biotic interactions [29]. In this study, MaxEnt analysis results indicated that temperature and precipitation jointly limited the geographical distribution pattern of *C. gilva*. In the eight evaluation approaches used in this study, temperature factors consistently ranked high in importance, being the top factor four times and the second factor four times. Similarly, precipitation factors were the top or second factor four times each. From this perspective, temperature and precipitation factors exhibited equal importance in influencing the geographic distribution of *C. gilva*. However, in terms of contribution rates and permutation importance values, temperature factors accounted for 72.6% and 83.4%, while precipitation factors only contributed 27.4% and 16.6%. Overall, temperature factors had a more significant impact on the distribution pattern of *C. gilva* compared with precipitation factors. This contrasts with a study by Zhang Lijuan, which indicated that annual precipitation (Bio12) was the most crucial factor limiting the geographic distribution of another species in the *Quercus* genus, *Quercus glauca*, followed by temperature annual range (Bio7) and mean temperature of coldest quarter (Bio11) [30]. MaxEnt predictions showed that the mean diurnal mean range (Bio2) had the highest contribution rate and regularization training gain when used alone, while the permutation importance of min temperature of coldest month (Bio6) was the highest. Jackknife tests also suggested that temperature factors were essential in constraining the potential geographic distribution of *C. gilva*. Existing research has indicated that temperature factors play an irreplaceable role in shaping the distribution patterns of plant species [31,32].

Changes in the geographic distribution of plants are closely related to climate change, which affects the potential distribution of plants by influencing key factors such as temperature and precipitation. Other species in the *Quercus* genus, such as *Cyclobalanopsis glauca* and *Cyclobalanopsis glaucoides*, are also constrained by temperature and precipitation factors [33]. In this study, comprehensive analyses using contribution rates, permutation importance values, and Jackknife tests showed that the mean diurnal mean range (Bio2), min temperature of coldest month (Bio6), precipitation of driest quarter (Bio17), and precipitation of driest month (Bio14) played crucial roles in shaping the geographic distribution pattern of *C. gilva*. Specifically, mean diurnal range (Bio2) had the highest contribution rate and regularization training gain, with a predicted suitable range of 3.92 to 9.30 °C for *C. gilva*. The asymmetric warming trend caused by different diurnal temperature ranges may impact carbon absorption and consumption in plants [34]. Photosynthesis and transpiration in plants predominantly transpire during daylight hours, exhibiting heightened sensitivity to diurnal temperature elevations. Conversely, the process of respiration is a continuous phenomenon, rendering both diurnal and nocturnal temperature increments influential on plant [35]. The asymmetric warming trend, where nighttime temperatures rise faster than daytime temperatures, may not favor the growth of plants [36]. Plant growth has specific temperature thresholds, and the adaptation rate may struggle to keep up with the rate of warming, ultimately leading to inhibited vegetation growth or even death [37]. Therefore, it is speculated that the mean diurnal range of temperature is a significant factor limiting the extensive distribution of *C. gilva*.

The MaxEnt simulation results also indicated that min temperature of coldest month (Bio6) had the highest permutation importance value. The bud period of *C. gilva* coincides with winter, and the low temperatures during winter limit seed germination or seedling growth. Harsh cold conditions pose a severe challenge, preventing *C. gilva* from completing its life cycle in northern regions. The predicted suitable range for the coldest month’s minimum temperature in this study was −5.15 to 15.05 °C, and most northern regions experience winter temperatures below this lower limit. Therefore, the low temperatures in northern winters may be a primary factor restricting the northward expansion of *C. gilva*, further emphasizing the dominant role of temperature factors in limiting its distribution.

Additionally, the MaxEnt simulation results suggested that precipitation of driest quarter (Bio17) and precipitation of driest month (Bio14) were also crucial driving factors shaping the geographic distribution pattern of *C. gilva*. Driest month precipitation had the highest test gain and area under the receiver operating characteristic curve (AUC) when used alone, as it contained more effective information. Driest quarter precipitation exhibited the greatest reduction in regularization training gain and AUC when other bioclimatic variables were used, indicating that it contained information not present in other variables. Moreover, in this study, driest month precipitation and driest quarter precipitation were highly correlated (correlation coefficient R = 0.990), representing the maximum tolerance of plants to drought stress. The research of Wu Lijun [38] indicates that drought stress plays a critical role in the growth and development of *C. gilva*. The total biomass increment of *C. gilva* seedlings show a decreasing trend with the degree of drought stress. In summary, climate factors shape the geographic distribution pattern of *C. gilva* by influencing its growth and development, with temperature and precipitation factors playing a dominant role.

Additionally, this study employed the MaxEnt model to focus on analyzing the impact of climate change on the geographical distribution of *C. gilva*. However, we also recognize that, in addition to climatic factors, disturbance dynamics (such as fires and pests) and non-climatic factors like soil and surface sediments also significantly influence species distribution. Although we intended to incorporate these factors into the model, there is currently a relative scarcity of specific data for the Last Interglacial, Last Glacial Maximum, and mid-Holocene, which may potentially limit the completeness of our model. We recommend that future research should aim to fill this data gap by collecting and integrating environmental data from key periods through interdisciplinary collaboration. Furthermore, as new data are obtained, the model should be iteratively updated to enhance its predictive accuracy and ecological interpretability, thereby providing a more solid scientific foundation for the ecological conservation of *C. gilva* and other species.

Our research findings distinctly indicate that the optimal growth areas for *C. gilva* are primarily located in the southeastern part of China, specifically in the eastern region of Fujian Province and the central-eastern part of Taiwan. The climatic characteristics of these areas are highly consistent with the ecological requirements of the species. Specific temperature and precipitation patterns in these regions, including mean diurnal range (Bio2), min temperature of coldest month (Bio6), precipitation of driest quarter (Bio17), and precipitation of driest month (Bio14), provide ideal conditions for the growth of *C. gilva*. Furthermore, our study conducted an analysis of the bioclimatic features of other regions in southeastern China, revealing significant similarities with the highly suitable growth areas for *C. gilva* in terms of bioclimatic conditions, encompassing comparable precipitation and temperature patterns. The presence of *C. gilva* in these areas, as reported in the literature, suggests their potential habitats for the species [39,40]. By integrating these analyses, we have gained a deeper understanding of the bioclimatic needs and geographical distribution of *C. gilva*. This understanding is crucial for devising effective conservation and management strategies to ensure the survival and propagation of the species under current and future climate change scenarios. Identifying and assessing potential areas with similar bioclimatic conditions allows us to offer more precise and targeted recommendations for the conservation efforts of *C. gilva*.

### 4.3. Patterns of Potential Distribution Changes

MaxEnt predictions for the highly suitable areas of *C. gilva* during the the Last Interglacial Period (LIG) primarily concentrated in eastern Zhejiang, northeastern Fujian, and the central-eastern part of Taiwan. Sporadic distributions were also observed in areas such as Mount Yandang, Mount Wuyi, and Mount Daiyun. The highly suitable areas exhibited an isolated and fragmented distribution pattern. The *Quercus arbutifolia*, a member of the genus *Quercus*, has experienced evolutionary changes since the late Miocene epoch, suggesting the presence of the *Quercus* genus in southeastern China during that period [41]. Therefore, these regions might have served as highly suitable areas for *C. gilva* during the LIG. This study selected climate data from 1970 to 2000 as the baseline, as the climatic characteristics of this period coincide with the current ecological adaptability of *C. gilva*, providing a practical starting point for assessing its adaptability to climate change. Compared with historical interglacial periods, the recent climatic conditions are more relevant to current ecological conservation needs, thereby aiding us in more accurately predicting the potential impacts of climate change on species distribution and formulating effective conservation strategies accordingly.

By the Late Glacial Maximum, both the total and highly suitable areas of *C. gilva* reached their maximum extent. Many areas within mainland China, such as the central part of Jiangxi and the southern part of Hunan, transitioned from moderate to low suitable areas to highly suitable areas. Pollen analysis of the inland areas in the northern part of the South China Sea region from the Late Glacial Maximum and early Holocene revealed the presence of a genus closely related to *Cyclobalanopsis* [42]. Pollen analysis also indicated an increase in the abundance of *Quercus* genus in the southern mountainous region of Jiangxi as the environment warmed after the last glacial period, consistent with the results of this study [43]. Parts of the East China Sea shelf, the Tsushima Strait, and the southern islands of Japan became suitable areas, with some areas showing highly suitable zones. Studies suggest that, during the Late Glacial Maximum, the East China Sea experienced a sea level drop of approximately 120–140 m, resulting in significant exposure of the continental shelf [44]. Zheng Zhuo’s research indicates that the continental shelf of the East China Sea during the Late Glacial Maximum may have been a vast area dominated by the Yangtze River delta, with widespread plains, grasslands, and hidden wetlands [45]. In mountainous regions, there was a distribution of subtropical broad-leaved forests [46]. Ancient vegetation analysis in Japan revealed the development of deciduous broad-leaved forests, including oaks, around Lake Biwa’s eastern shore Sone Pond during the period 17,000–15,000 cal BP [47]. Therefore, the East China Sea continental shelf and the southern regions of the Japanese archipelago might have served as highly suitable areas for *C. gilva* during the Late Glacial Maximum.

The transition from the Late Glacial Maximum to the mid-Holocene was marked by a succession of deglaciation and warming. The climate tended to be warmer and more humid, leading to the transformation of forest vegetation types in southeastern China from deciduous broad-leaved forests to evergreen broad-leaved forests [46]. Additionally, pollen studies by Shuaili Li found the presence of subtropical broad-leaved tree species, including *Quercus* genus, in the Ningde region of Fujian during the mid-Holocene [48]. Therefore, *C. gilva* likely had a widespread distribution during the mid-Holocene.

In future climate change scenarios represented by ssp126 and ssp585, which, respectively, indicated the lowest and highest greenhouse gas concentration emission scenarios, the analysis of *C. gilva’* geographic distribution pattern revealed significant information.

From the present to the 2050s and then to the 2090s, the total suitable area showed an initial increase followed by a contraction trend under both climate scenarios. Except for the 2090s’ ssp585 scenario, the total suitable area of *C. gilva* expanded, compared with the contemporary period under the other three climate scenarios. This suggests that future climate change, driven by increasing surface temperatures and human activities contributing to a substantial rise in greenhouse gas concentrations, could initially promote photosynthesis in plants, benefiting their growth and development [49]. However, if temperatures continue to rise, the opposite effect may occur, negatively impacting plant growth [50]. In future climate scenarios, the highly suitable areas of *C. gilva* did not exhibit a clear pattern, showing an overall significant contraction trend. Particularly in the 2090s’ ssp585 scenario, the contraction of the highly suitable areas was severe, reducing by as much as 67.5% compared with the contemporary period. This trend suggests that future climate change may significantly impact the highly suitable areas of *C. gilva*.

From the present to the 2050s and then to the 2090s, under the ssp126 scenario, the MaxEnt model indicated a migration trend of the centroid in Jiangxi Province, southeast China, first moving northeast and then southeast, with migration distances of 36.21 km and 62.01 km, respectively. Under the ssp585 scenario, the predicted results showed a larger migration trend of the centroid in Jiangxi Province, initially moving east and then southwest, with migration distances of 93.52 km and 127.04 km, respectively. However, the results of Ouyang Zeyi’s study on the future suitable areas of *C. gilva* showed an overall trend of migrating toward higher latitudes and different periods under various climate scenarios, inconsistent with the results of this study [25]. Interestingly, the results of this study showed that, from the Late Glacial Maximum to future climate scenarios, the centroid of *C. gilva* distribution remained in the southeastern part of China, specifically within Jiangxi Province. We speculate that, except during the Late Glacial Maximum, when the suitable area of *C. gilva* experienced significant expansion, the contour range of the suitable area was generally similar during other periods, with no large-scale expansion or contraction. In the current context of global warming, more studies indicate that plants are migrating to higher latitudes [4].

### 4.4. Recommendations for the Protection of Cyclobalanopsis gilva

In areas where the suitable habitat is preserved, on-site protection measures can be implemented based on the natural distribution of *C. gilva*, along with the formulation of corresponding conservation policies. Our team’s extensive field surveys have identified priority conservation areas for *C. gilva* in several locations (Figure 6), including Qingtianyan Village in Longcun Township, Jianou City, Fujian; Jihui Village in Lidun Town, Zhounan County, Fujian; Songyang City, Zhejiang; Taozilou Village in Bichong Town, Zhijiang, Hunan; Bako Village in Maogou Town, Baojing County, Hunan; Tuanzhai Bian in Xikou Town, Tongdao County, Hunan; Chaliu in Suining County, Hunan; Chadang Village in Suien County, Jiangxi; and Xinzai Township, Changshun County, Guizhou. For known populations of *C. gilva*, urgent measures are needed to establish on-site protection zones in these areas. Through targeted observation, dynamic monitoring, analysis, and strengthened management, changes in population and habitat can be determined, contributing to the protection of genetic resources.

It is noteworthy that the majority of Hunan Province and the eastern part of Fujian Province consistently remain in the moderately suitable zone across different periods. These regions could serve as refuges for *C. gilva* in response to future climate change.

In cases where suitable habitats are lost, proactive measures such as translocation should be adopted. Establishing botanical gardens can facilitate the transplantation of *C. gilva* into artificial environments for cultivation, maintenance, and preservation. Given its preference for mountainous regions at altitudes ranging from 300 to 1500 m, selective creation of artificial *C. gilva* communities should be undertaken. Translocation efforts should involve cultivating seedlings through transplanting large seedlings, seed germination, and asexual reproduction, with the aim of maintaining viable population numbers to preserve genetic diversity. Genetic resource repositories have already been established in places like Yuchi State-owned Forest Farm in Miluo City, Hunan, and Anyuan Experimental Forest Farm in Jiangxi.

Currently, the availability of high-quality *C. gilva* seeds is limited, and the area under artificial cultivation is relatively small, leading to low yields of usable materials. This restricts the realization of its potential as a valuable timber resource. Therefore, we recommend expanding afforestation efforts in areas of moderate to high suitability for *C. gilva*, alleviating the current scarcity of this species.

## 5. Conclusions

This study, utilizing an optimized MaxEnt model, simulated the geographical distribution patterns of *C. gilva* in response to climate change under different periods and scenarios. The suitable habitat for *C. gilva* is primarily located in the southeastern regions of China, with highly suitable areas concentrated in the eastern part of Fujian and the central-eastern part of Taiwan. Temperature and precipitation jointly influence the geographic distribution of *C. gilva*, with temperature playing a dominant role. Variables such as mean diurnal range (Bio2), min temperature of coldest month (Bio6), precipitation of driest quarter (Bio17), and precipitation of driest month (Bio14) are identified as key bioclimatic variables restricting the distribution of *C. gilva*. This study provides valuable insights into the response of *C. gilva* to future climate change scenarios and offers recommendations for its conservation and sustainable management.

## Figures and Tables

**Figure 1 plants-13-02336-f001:**
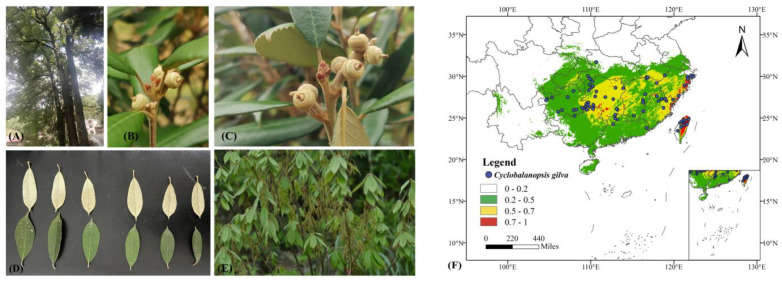
Photos and distribution points of *Cyclobalanopsis gilva*. (**A**) Whole tree; (**B**,**C**) fruits; (**D**) leaves; (**E**) flowers; (**F**) distribution points.

**Figure 2 plants-13-02336-f002:**
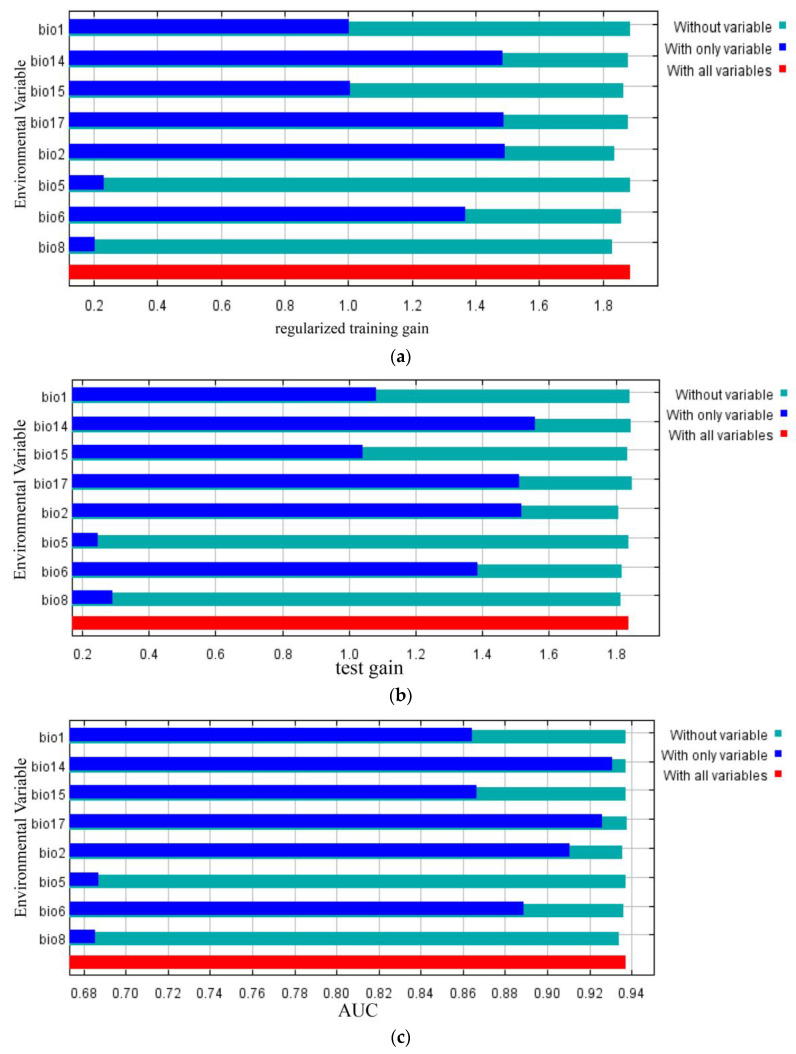
Jackknife test of the importance of variables. Blue, green, and red bars represent running the MaxEnt model with the variable alone, without the variable, and with all variables, respectively (**a**) regularized training gain; (**b**) test gain; (**c**) The area under the ROC curve (AUC).

**Figure 3 plants-13-02336-f003:**
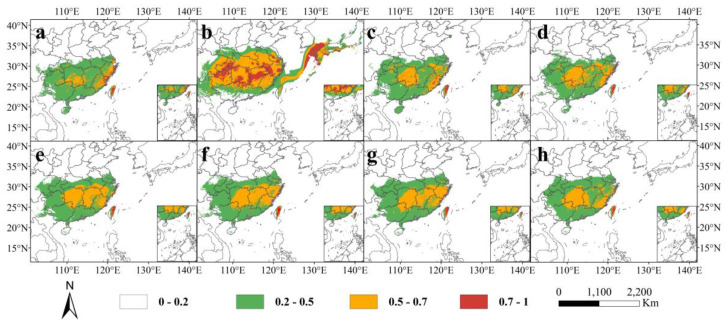
Predicted suitable distribution areas of *C. gilva* based on MaxEnt model: (**a**) Last Interglacial (LIG), (**b**) Last Glacial Maximum (LGM), (**c**) Middle Holocene (MH), (**d**) current, (**e**) representative concentration pathways 2050 s-SSP126, (**f**) 2050 s-SSP585, (**g**) 2090 s-SSP126, and (**h**) 2090 s-SSP585.

**Figure 4 plants-13-02336-f004:**
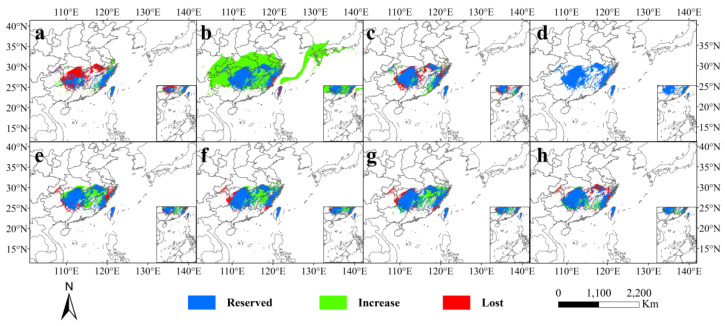
Spatial transformation pattern of suitable areas for *C. gilva* in different periods. (**a**) Last Interglacial (LIG), (**b**) Last Glacial Maximum (LGM), (**c**) Middle Holocene (MH), (**d**) current, (**e**) representative concentration pathways 2050 s-SSP126, (**f**) 2050 s-SSP585, (**g**) 2090 s-SSP126, and (**h**) 2090 s-SSP585.

**Figure 5 plants-13-02336-f005:**
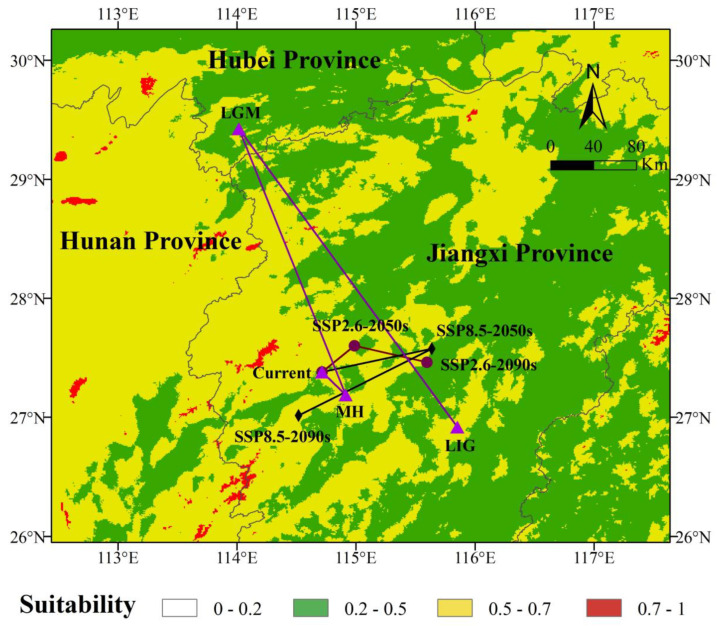
Migration location of the center of suitable areas of *C. gilva* under different climate change scenarios.

**Figure 6 plants-13-02336-f006:**
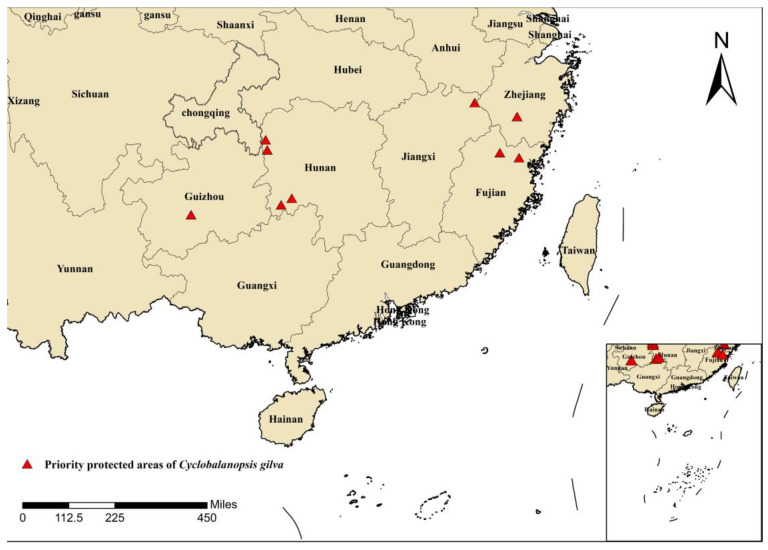
Priority protected areas for *C. gilva*.

**Table 1 plants-13-02336-t001:** Evaluation results of MaxEnt model under different parameter settings.

Model Evaluation	Feature Combination	Regularization Multiplier	Delta. AICc	Avg. diff. AUC	Avg. Test. or10pct
default	LQHPT	1	80.56426871	0.023329891	0.152252568
optimize	LQ	1.8	0	0.011517256	0.141835901

**Table 2 plants-13-02336-t002:** Bioclimatic variables and their main parameters of *C. gilva*.

Code	Bioclimatic Variables	PC (%)	PI (%)	TRG_W_	TRG_O_	TG_W_	TG_O_	AUC_W_	AUC_O_
Bio2	Mean Diurnal Range (mean of monthly (max temp–min temp))	40.0	19.7	1.8357	1.4913	1.8069	1.5168	0.9356	0.9109
Bio6	Min Temperature of Coldest Month	28.1	54.1	1.8591	1.3691	1.8187	1.3867	0.9362	0.8890
Bio17	Precipitation of Driest Quarter	16.9	1.7	1.8782	1.4877	1.8479	1.5112	0.9377	0.9263
Bio15	Precipitation Seasonality (Coefficient of Variation)	5.6	12.8	1.8658	1.0065	1.8349	1.0400	0.9372	0.8665
Bio14	Precipitation of Driest Month	4.9	2.1	1.8812	1.4860	1.8448	1.5578	0.9374	0.9312
Bio8	Mean Temperature of Wettest Quarter	3.3	8.7	1.8311	0.2048	1.8132	0.2921	0.9339	0.6855
Bio1	Annual Mean Temperature	1.0	0.4	1.8856	1.0031	1.8401	1.0828	0.9371	0.8646
Bio5	Max Temperature of Warmest Month	0.2	0.5	1.8854	0.2342	1.8398	0.2478	0.9372	0.6870

Note: PC is percent contribution; PI is permutation importance; TRG_O_ is the regularization training gain using the factor alone; TRGW is the regularization training gain using other factors; TG_O_ is the test gain using the factor alone; TGw is the test gain using other factors; AUCw is the area under the receiver operating characteristic curve using other factors; AUCo is the area under the working characteristic curve of the subjects using the variable alone.

**Table 3 plants-13-02336-t003:** Changes in suitable areas of *Z. serrata* in different periods (unit: ×10^4^ km^2^). Values in parenthesis represent proportion of areas (%).

Period	Generally Suitable Habitats	Moderately Suitable Habitats	Highly Suitable Habitats	Total
Last Interglacial	118.21	24.6	3.87	146.68
Last Glacial Maximum	180.54	118.66	46.52	345.72
Middle Holocene	107.68	37.07	2.92	147.67
Current	99.5	40.41	3.14	143.05
2050 s-SSP126	115.35	50.41	1.27	167.03
2050 s-SSP585	112.3	50.11	1.6	164.01
2090 s-SSP126	110.65	48.29	1.66	160.6
2090 s-SSP585	113.04	40.75	1.02	154.81

**Table 4 plants-13-02336-t004:** Changes in suitable areas of *C. gilva* in different periods.

Period	Area (km^2^)				Change (%)			
	Increase	Reserved	Lost	Change	Increase Rate	Reserved Rate	Lost Rate	Change Rate
Last Interglacial	6.12	22.22	21.27	−15.16	14.05%	51.03%	49.85%	−34.81%
Last Glacial Maximum	124.42	40.64	2.88	121.54	285.70%	93.33%	6.61%	279.10%
Middle Holocene	7.47	32.38	11.18	−3.71	17.16%	74.37%	25.68%	−8.52%
2050 s-SSP126	14.90	36.66	6.70	8.19	34.21%	84.17%	15.39%	18.82%
2050 s-SSP585	14.24	37.49	5.97	8.28	32.70%	86.10%	13.70%	19.01%
2090 s-SSP126	12.57	37.19	6.26	6.31	28.86%	85.40%	14.36%	14.49%
2090 s-SSP585	6.81	35.02	8.40	−1.59	15.64%	80.41%	19.29%	−3.65%

(unit: ×10^4^ km^2^).

## Data Availability

All data necessary to replicate this study’s results are included in this published article (and its Appendix A). Raw data are available upon request.

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
