# Peer review of "The Potential Habitat Response of Cyclobalanopsis gilva to Climate Change"

_plants, 2024, doi:10.3390/plants13162336_

Round 1

Reviewer 1 Report

Comments and Suggestions for Authors

This work studied the impacts of potential changes in climate on habitat expansion or contraction of Quercus gilva. The authors projected Q. gilva distribution during the last inter-glacial and the last glacial maximum and future years. I enjoyed reading the work, it is nicely written and it has the potential to make great contribution to understanding climate change impacts on landscape level vegetation dynamics. Authors need to clarify and tidy a few aspects of this work;

1) Cyclobalanopsis gilva distribution points;

- In the Figure 1, the distribution data is not a presence/absence kind of data but rather the proportion or percent cover of C. gilva for each point, is that correct? If yes, please state this explicitly in the paper. Is this the number of C. gilva stems per hectare as percentage or basal area proportion etc. 

- Could authors add a column to Table S1 that shows which percent cover class each point belongs to, this will help understand the distribution numerically.

- In order to avoid clustering effects, authors retained only data points that allowed for maintaining a 10 km distance between points. how does this impact the results since authors are modelling distribution but in this case avoiding clustering??

2) Environmental Variables;

- Is there any a particular reason why the period 1800 to 1969 was excluded from the analysis?

- What is the spatial resolution of the BCC-CSM2-MR model? 

- for all the environmental data, can we have a sense of how much they vary i.e. mean, min, max as a supplementary file?

- Pearson correlation table (Table S3): bio14 is not in the Table and bio16 is not defined in the text 

- there is significantly high correlation between some of the variables e.g. bio1 & bio5, bio1 & bio6, bio1 & bio7 but most importantly bio6 & bio8 because these pair are retained in the final model. How does the multicolinearity here impact the 97.4% explained variance?

- How different are temperature changes as you move from last interglacial to last interglacial maximum to holocene?

3) Habitat suitability;

- how was habitat suitability determined, please explain the "natural break method: and define the P in "0≤P<0.2". Can this P data be added as a column to the Table S1?

4) Spatial distribution pattern changes;

- Are the projections present/absent data or they are also percent data like the input data?

- Please mention somewhere in the methods that the contemporary period (1970 - 2000) is the "reference" or "baseline" for calculating changes? 

- And please justify why this is the reference and for example the interglacial period is not the reference

- How was centroid calculated?

- Can the authors map the recommended protection areas listed at L500-507

5) Other potential drivers of vegetation changes;

- Since C. gilva is sensitive to altitude, why is this variable not included in the modeling?

- Another potential limitation is that no information on disturbance dynamics (fire, pests etc.), soil/surficial deposit are included in the model, why?

Minor comments;

- Please stick with either only Cyclobalanopsis gilva or Quercus gilva as using the two inter-changeably can confuse the reader

Abstract section: please remove bio2, bio6 etc. from the abstract section

Table 2; please write the environmental variables in full

Table 4; please write periods in full

Reviewer 2 Report

Comments and Suggestions for Authors

The study deals with the distribution of Cyclobalanopsis gilva for its timber interest in China, and the influence of climate change on the distribution of this species. The introduction is correct because it deals with the background of this species. his species belongs to the family Fagaceae, however in Figure 1 where the authors give 73 points of natural distribution, the name Quercus gilva is written. The authors should opt for one of the two names to add authorships, always taking into account the prioritization of the name according to the International Code of Botanical Nomenclature.

The data on environmental variables used are correct, but they are exclusively climatic data. P In order to optimize the proposed model, I advise the authors to incorporate a bioclimatic analysis of the most optimal areas for this species, and a bioclimatic analysis of other potential areas with the same bioclimate

I note that Figures 3 and 4 do not distinguish the patches well, so I advise improving these maps. The variables bio1 to bio17 are exclusively physical, please change the acronyms to avoid confusion.

Round 2

Reviewer 1 Report

Comments and Suggestions for Authors

The authors provided a thorough response to my reviews. Here are a few follow up recommendations;

1) "The geographical distribution data utilized in this study are primarily sourced from recent records of herbaria" Please specify the time period of the observed distribution data points.

2) Only climate data over the period 1970-2000 was used in calibrating the model because this was the only contemporary climate data available in WorldClim 2.1. I wonder if this should be mentioned somewhere in the discussion section as potential limitation of the study?

3) Thank you for adding Table S4. The change in C. gilva distribution from LIG to LGM to the Holocene is quite high (Table 3, Figure 3). It would be helpful if authors can expand the Table S4 to show the average climate variables for LIG, LGM and Holocene. 

4) Authors could add the "Response 14" to the manuscript which justifies using 1970-200 as baseline. 

5) Please add the explanation of how centroid was calculated to the methods section (Response 15)

6) Please add explanation of the lack of data on disturbances for the LIG, LGM and Holocene as basis for not including these in the model to the discussion section or as recommendation for future improvement to the work.

Author Response

请参阅附件。
